# In situ structural analysis of the *Yersinia enterocolitica* injectisome

**Mikhail Kudryashev[1†], Marco Stenta[2†], Stefan Schmelz[3†], Marlise Amstutz[4†], Ulrich Wiesand[3,4†], Daniel Castaño-Díez[1], Matteo T Degiacomi[2], Stefan Münnich[3], Christopher KE Bleck[1], Julia Kowal[1], Andreas Diepold[4‡], Dirk W Heinz[3*], Matteo Dal Peraro[2,5*], Guy R Cornelis[4*], Henning Stahlberg[1*]**

[1]Center for Cellular Imaging and NanoAnalytics (C-CINA), Biozentrum, University of Basel, Basel, Switzerland; [2]Institute of Bioengineering, School of Life Sciences, École Polytechnique Fédérale de Lausanne (EPFL), Lausanne, Switzerland; [3]Division of Structural Biology, Helmholtz Centre for Infection Research, Braunschweig, Germany; [4]Focal Area Infection Biology, Biozentrum, University Basel, Basel, Switzerland; [5]Swiss Institute of Bioinformatics (SIB), Lausanne, Switzerland

**Abstract** Injectisomes are multi-protein transmembrane machines allowing pathogenic bacteria to inject effector proteins into eukaryotic host cells, a process called type III secretion. Here we present the first three-dimensional structure of *Yersinia enterocolitica* and *Shigella flexneri* injectisomes in situ and the first structural analysis of the *Yersinia* injectisome. Unexpectedly, basal bodies of injectisomes inside the bacterial cells showed length variations of 20%. The in situ structures of the *Y. enterocolitica* and *S. flexneri* injectisomes had similar dimensions and were significantly longer than the isolated structures of related injectisomes. The crystal structure of the inner membrane injectisome component YscD appeared elongated compared to a homologous protein, and molecular dynamics simulations documented its elongation elasticity. The ring-shaped secretin YscC at the outer membrane was stretched by 30–40% in situ, compared to its isolated liposome-embedded conformation. We suggest that elasticity is critical for some two-membrane spanning protein complexes to cope with variations in the intermembrane distance.

**\*For correspondence:** Dirk. Heinz@Helmholtz-HZI.de (DWH); Matteo.Dalperaro@epfl. ch (MDP); Guy.Cornelis@fundp. ac.be (GRC); Henning. Stahlberg@unibas.ch (HS)

†These authors contributed equally to this work

‡**Present address:** Department of Biochemistry, University of Oxford, Oxford, United Kingdom

**Reviewing editor**: Volker Dötsch, Goethe University, Germany

## Introduction

The bacterial type III secretion apparatus, the injectisome, is a complex nanomachine that allows Gram-negative bacteria to export effector proteins in one step across the two bacterial membranes and an eukaryotic cell membrane (*Cornelis, 2006*; *Galan and Wolf-Watz, 2006*). The assembly of the injectisome involves some 34 different proteins. Many of these proteins form the structure, while others act as ancillary components driving the assembly process. Phylogenic analyses based on the most conserved proteins classify injectisomes into seven different families (Ysc, Ssa-Esc, Inv-Mxi-Spa, Hrc1, Hrc2, Rhizobiales, and Chlamydiales) (*Pallen et al., 2005*; *Troisfontaines and Cornelis, 2005*). The injectisome consists of three parts: a ~60 nm long, hollow needle protruding from the bacterial surface, a basal body that spans the two bacterial membranes and the periplasm, and a cytoplasmic part. Recent atomic models allowed a mechanistic understanding of the needle structure (*Fujii et al., 2012*; *Loquet et al., 2012*).

The basal body of the *Salmonella enterica* serovar Typhimurium SPI-I and *Shigella flexneri* injectisomes have been purified and structurally analyzed in great detail. The basal body presents a barrel-shaped structure at the outer membrane (OM), with double ring-shaped densities underneath in the periplasmic space and the inner membrane (IM). These rings are formed by three multimeric proteins (YscC,D,J in *Yersinia spp*; InvG, PrgH,K in *Salmonella* SPI-1; MxiD,J,G in *S. flexneri*). The barrel-shaped structure spanning the OM and protruding into the periplasm consists of a 12–15 mer of a protein from the secretin

**eLife digest** Humans and other animals can use the five senses—touch, taste, sight, smell, and hearing—to interpret the world around them. Single-celled organisms, however, must rely on molecular cues to understand their immediate surroundings. In particular, bacteria gather information about external conditions, including potential hosts nearby, by secreting protein sensors that can relay messages back to the cell.

Bacteria export these sensors via secretion systems that enable the organism both to receive information about the environment and to invade a host cell. A total of seven separate secretion systems, known as types I–VII, have been identified. These different secretion systems handle distinct cargoes, allowing the bacterial cell to respond to a range of feedback from the external milieu.

The type III secretion system, also known as the 'injectisome', is found in bacterial species that are enclosed by two membranes separated by a periplasmic space. The injectisome comprises different components that combine to form the basal body, which spans the inner and outer membranes, and a projection from the basal body, called the hollow needle, that mediates the export of cargo from a bacterium to its host or the local environment.

The distance between the inner and outer membranes may vary across species or according to environmental conditions, so the basal body must be able to accommodate these changes. However, no mechanism has yet been established that might introduce such elasticity into the injectisome. Now, Kudryashev et al. have generated three-dimensional structures for the injectisomes of two species of bacteria, *Shigella flexneri* and *Yersinia enterocolitica*, and shown that the size of the basal body can fluctuate by up to 20%.

Kudryashev et al. imaged whole injectisomes in these two species and found that the height of the basal body was proportional to the distance between the inner and outer membranes. To probe how this could occur, the properties of two proteins that are important components of the basal body were studied in greater detail. YscD, a protein that extends across the periplasmic space, was crystallized and its structure was then determined and used to develop a computer model to assess its compressibility: this model indicated that YscD could stretch or contract by up to 50% of its total length. The outer membrane component YscC also appeared elastic: when the protein was isolated and introduced into synthetic membranes, its length was reduced 30–40% relative to that observed in intact bacterial membranes.

A further experiment confirmed the adaptability of the basal body: when the separation of the membranes was deliberately increased by placing bacteria in a high-salt medium, the basal body extended approximately 10% in length. Cumulatively, therefore, these experiments suggest that the in-built flexibility of the basal body of the injectisome allows bacteria to adjust to environmental changes while maintaining their sensory abilities and host-invasion potential.

family (YscC, InvG, MxiD) (**Burghout et al., 2004**; **Marlovits et al., 2004**; **Hodgkinson et al., 2009**; **Spreter et al., 2009**; **Schraidt and Marlovits, 2011**). The lower double-ring reaching into the IM is made of a lipoprotein (YscJ, PrgK, MxiJ) proposed to form a 24-subunit ring (**Kimbrough and Miller, 2000**; **Crepin et al., 2005**; **Yip et al., 2005**; **Silva-Herzog et al., 2008**; **Hodgkinson et al., 2009**) and a protein that is not so conserved as the others, but shares a similar modular domain architecture and acts as a connector between the secretin and the IM (YscD, PrgH, MxiG) (**Spreter et al., 2009**; **Diepold et al., 2010**). The injectisome is evolutionarily related to the bacterial flagellar motor system, with which it shares the basic type III secretion export apparatus (**Cornelis, 2006**; **Minamino et al., 2008**; **Erhardt et al., 2010**). This allows parallels to be drawn for the cytoplasmic elements of the injectisome (**Cornelis, 2006**; **Swietnicki et al., 2011**). The recent crystal structure of the C-terminus of the export gate of the cytoplasmic export apparatus MxiA from *S. flexneri* revealed a ring arrangement and allowed a mechanistic insight into secretion control (**Abrusci et al., 2013**).

Here we report the three-dimensional structure of the *Yersinia enterocolitica* and *S. flexneri* injectisomes in their native environment, that is, in the membrane of the intact bacterium. At the same time we provide structural information for the intact *Y. enterocolitica* injectisome, which so far has not been purified. Importantly, we document length adaptations of the injectisome basal body to variations in the distance between the bacterial inner and outer membranes. Using our crystallographic data on

YscD and molecular simulation and modeling with restraints from our cryo-EM map, we provide an atomistic model of the basal body YscDJ ring inserted in the bacterial inner membrane.

## Results

### Structure and in situ flexibility of the basal body

We applied cryo electron tomography (cryo-ET) to visualize injectisomes in intact *Yersinia enterocolitica* cells at nanometer-scale resolution and found variations in their appearance (*Figure 1*), indicating an intrinsic flexibility. Since thick cells reduced the quality of tomographic imaging, we genetically engineered *Y. enterocolitica* minicells that were thinner and still had assembled injectisomes (*Figure 1—figure supplement 1*). Reduced thickness allowed for imaging conditions permitting a higher resolution reconstruction (*Kudryashev et al., 2012a*). Sub-tomogram averaging of volumes containing injectisomes from tomograms of *Y. enterocolitica* minicells and focal pair tomograms (*Kudryashev et al., 2012b*) of wild type, regularly sized cells were performed using a customized local feature alignment strategy to compensate for structural variations among individual injectisomes (*Figure 2—figure supplement 1*, 'Materials and methods' for details). This yielded a reconstruction of the injectisome in situ at ~4 nm resolution (*Figure 2A*). The shape of the average structure is similar to what is expected from the current structural model (*Worrall et al., 2011*); the main components of the basal body were assigned as YscC and YscDJ rings following this nomenclature (*Figure 2A*) and the secretin YscC was assumed inserted into the outer membrane. YscD is known to have four periplasmic domains. The first three are thought to form a ring structure in analogy to homologues (*Spreter et al., 2009*), while the fourth domain mediates the interaction with YscC (*Ross and Plano, 2011*). The YscC-D

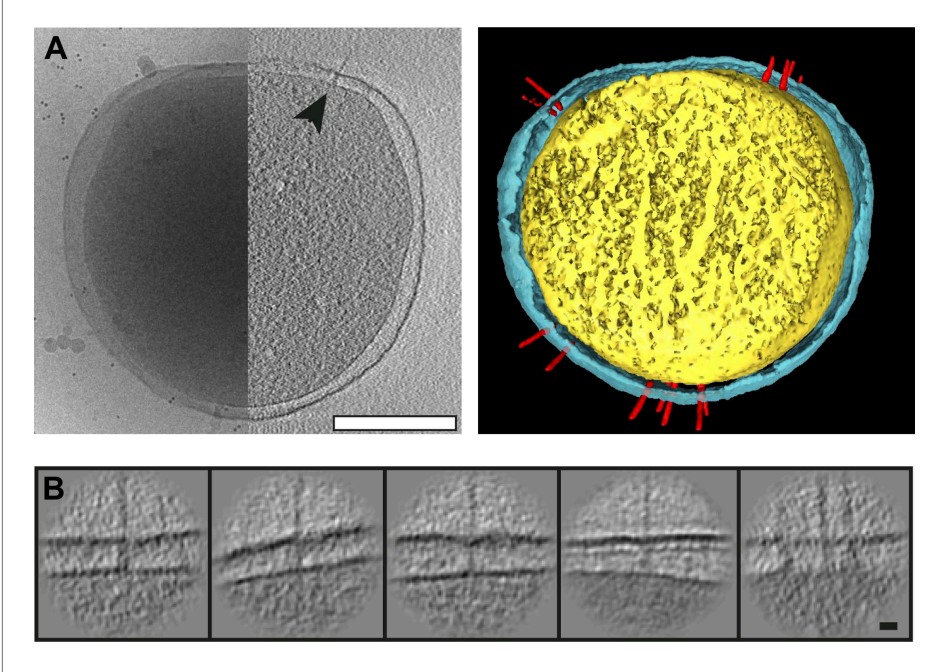

**Figure 1**. Visualization of *Y. enterocolitica* injectisomes in situ. (**A**) Left: cryo-EM image of a *Yersinia enterocolitica* bacteria (left half of the panel) and a 20-nm thick slice through a tomogram of the bacteria, showing an injectisome (right half of the panel, black arrowhead); Right: volume rendering of the same *Y. enterocolitica* bacteria showing the inner (yellow) and outer (blue) membranes, and injectisomes (red). (**B**) Example images of individual injectisomes, illustrating different types of observed injectisomes. Left to right: regular, tilted, with dim basal body, denser peptidoglycan layer, and clustered injectisomes. Scale bars: **A**: 300 nm, **B**: 20 nm.
The following figure supplements are available for figure 1:

**Figure supplement 1**. *Y. enterocolitica* minicells.

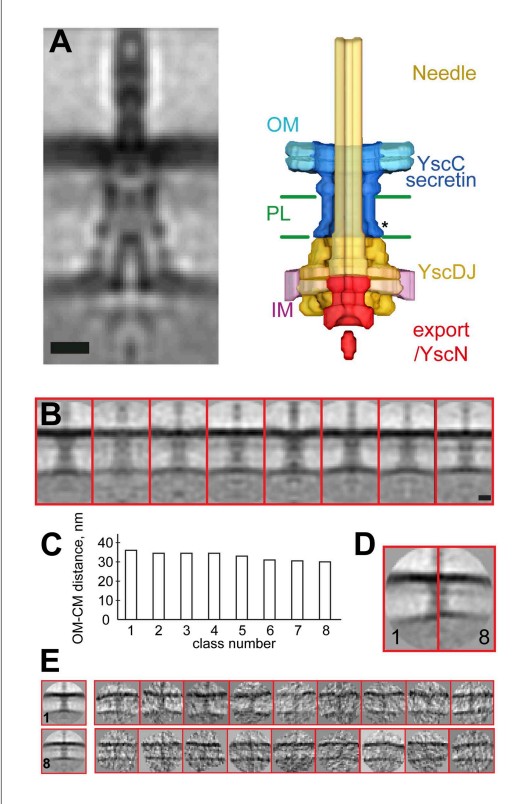

**Figure 2**. Structure of the *Y. enterocolitica* injectisome in situ. (**A**) Slice through the average 3D structure of the injectisome and a model with indicated components. OM–outer membrane, PL–peptidoglycan layers, IM–inner membrane, * indicates the junction between YscC and YscD. (**B**) 8 class averages of injectisomes from wt cells obtained by MRA classification; their length varies significantly. (**C**) Intermembrane distances for the corresponding class averages from (**B**), the longest class has a distance of 36 nm, the shortest of 30 nm. (**D**) Overlay of the longest and the shortest class aligned by the OM. (**E**) Class averages 1 (longest) and 8 (shortest), together with representative individual injectisomes for the two classes, all at the same scale. Scale bars: 10 nm; box heights for **B**,**D**,**E**: 96 nm.

The following figure supplements are available for figure 2:

**Figure supplement 1**. Structural elasticity of the injectisome.

**Figure supplement 2**. Comparison of membrane-to-membrane distance using CEMOVIS and cryo-ET of plunge frozen *Y. enterocolitica*.

junction is distinguishable in the density map and indicated in the density assignment (*Figure 2A*, asterisk).

The vertical distance between the centers of the membranes in the reconstruction is ~33 nm. This average value is in the range of values reported for gram-negative bacteria (*Chen et al., 2011*; *Liu et al., 2011*; *Wang et al., 2012*), but significantly larger than the ~20 nm expected from the structures of basal bodies isolated from *Salmonella enterica SPI-I* and *S. flexneri* (*Hodgkinson et al., 2009*; *Schraidt and Marlovits, 2011*). Independent measurements made by imaging *Y. enterocolitica* cells by cryo-electron microscopy of vitreous sections (CEMOVIS) (*Figure 2—figure supplement 2* and 'Materials and methods') and tomography on high pressure frozen and freeze substituted bacteria (data not shown) confirmed the intramembrane distance. We thus conclude that the measured distance between the IM and OM, and hence the dimensions of the injectisome basal bodies being longer than isolated single particle structures, are unlikely to be a consequence of the cryo sample preparation method employed.

Measuring from mass center to mass center, the largest lateral diameter of the periplasmic part of the average injectisome structure is 18 nm (*Figure 2—figure supplement 1E*). This region is close to the IM, where YscDJ is localized (*Worrall et al., 2011*). The largest diameter at the outer OM is 12 nm (*Figure 2—figure supplement 1E*), which is in good agreement with the dimensions of isolated and liposome reconstituted YscC complexes (see below). The channel of the injectisome's needle is resolved at the OM. Further, a large ring-like structure can be discerned on the cytoplasmic side of the IM (*Figure 2A*, yellow). It surrounds a smaller, torus-like structure localized ~5 nm underneath the membrane, which we tentatively propose to be the export gate YscV (*Figure 2A*, red), based on the localization of the export gate protein FlhA of the flagella motor (*Abrusci et al., 2013*). Similarly, in analogy to the flagellar motor system (*Chen et al., 2011*), the density below the export gate likely corresponds to the YscN ATPase.

Classification by iterative multi reference alignment (MRA) of individual in situ *Y. enterocolitica* injectisomes revealed large variations in the length of the complex in the intermembrane space; intermembrane distances ranged from 30 to 36 nm (*Figure 2B–C*), suggesting that the basal body of the injectisome is highly flexible in the cellular context. Individual injectisomes belonging to the classes with the longest and shortest lengths are reproduced in *Figure 2E*. Long and short classes had similar densities corresponding to the peptidoglycan layer.

## Tertiary structure elasticity of YscD subunits in the basal body

In order to investigate the origin of the unexpected basal body flexibility observed by cryo-ET, we determined the crystal structure of the first three periplasmic domains of YscD from *Y. enterocolitica* (residues 150–362). The 2.7 Å resolution structure of YscD$^{150–362}$ (*Supplementary file 1A*) showed three linearly arranged domains (*Figure 3A* and *Figure 3—figure supplement 1*). Each domain comprises an αββαβ-ring building motif, which superposes well with corresponding motifs in the three domains of the homologous PrgH from *Salmonella enterica* (*Spreter et al., 2009*) (*Figure 3—figure supplement 1B,C*). However, although sharing the same multi-domain architecture, YscD features an extended conformation, while the domains of PrgH are in a more compact arrangement. The YscD structure points to a high inter-domain flexibility, as already indicated by the high B-factor values of the third domain of *wt* YscD$^{150–362}$ (*Figure 3—figure supplement 1D* and *Supplementary file 1A*). Molecular dynamics (MD) simulations revealed that the connecting hinge (residues A282-N284) between the second and third periplasmic domains is mainly responsible for the high flexibility of YscD (*Figure 3—figure supplement 2*, 'Materials and methods'). Based on this observation, we engineered a point mutation (G283P) at the hinge region, in order to stabilize YscD in a less flexible conformation. Mutation of G283 to a proline resulted in a protein that was still functional for type III secretion in vivo (*Figure 3—figure supplement 3*). The 1.4 Å resolution crystal structure of YscD$^{150–347}$ G283P had a slightly more bent conformation in the hinge region, but otherwise showed the same extended domain organization as YscD$^{150–362}$*wt*: the first two domains superpose well with the *wt* structure, while the third domain is tilted by ~9° with respect to the first two (*Figure 3—figure supplement 1A*). In contrast to the *wt* structure, the refinement statistics for the less flexible YscD$^{150–347}$ G283P are in the expected range (e.g., R$_{work}$/R$_{free}$ = 18.9/22.6) (*Supplementary file 1A*). Comparison of the dynamic mobility of each structure corroborated this; the proline mutant has lower atomic B-factors and hence a more rigid, but still functional, structure (*Figure 3—figure supplement 3* and 'Materials and methods').

We then explored the intrinsic flexibility of the first, second, and third periplasmic domains of YscD by MD simulations, probing their ability to access multiple conformations. We found that the *wt* domains, as well as YscD$^{150–347}$ G283P, can be arranged in a compact form similar to the corresponding region of the YscD homologue PrgH from *Salmonella SPI-1* (*Figure 3A*). Moreover, the application of stretching forces in MD produced an almost barrier-free transition from the compact to the extended conformation, resulting in a domain arrangement similar to that observed by X-ray crystallography. After this point the forces start to increase due to partial unfolding of secondary structure (*Figure 3B*). Due to this tertiary structure elasticity, that is, the rearrangement of tertiary structure in response to mechanical force, the periplasmic domains of YscD can undergo a maximal elongation of about 3.5 nm (i.e., a 50% elongation compared to the compact conformation), while opposing the stretching with a force as small as 35 ± 15 pN per YscD monomer, a value that is comparable with that estimated for other highly stretchable multi-domain proteins (*Hsin and Schulten, 2011*). This suggests that the tertiary structural elasticity of the ring components of the periplasmic portion of YscD significantly contributes to the overall elasticity of the basal body of the injectisome in response to external forces.

## Structural model of the elongated YscDJ inner ring of the basal body

We constructed a pseudo-atomic model of the inner membrane part of the basal body by combining experimental data with molecular modeling and simulations, similar as to exercised previously to unveil the architecture of large macromolecular complexes (*Alber et al., 2007*; *Lasker et al., 2012*). Multimers of the YscD and the YscJ proteins were assembled into symmetrical 24-meric ring models using a newly developed optimization protocol based on swarm intelligence heuristic method (*Degiacomi and Dal Peraro, 2013*). For this, we employed spatial restraints from the in situ cryo-ET map, cross-linking data from the PrgH/PrgK homologues (*Sanowar et al., 2010*; *Schraidt et al., 2010*), homology models of YscJ based on EscJ, and the conformational ensemble defined by MD simulations of YscD. YscD model structure also includes the transmembrane (TM) domain and the N-terminal cytosolic part of YscD. It does not include the last 70 amino acids of YscD, which interact with the secretin YscC (*Spreter et al., 2009*; *Schraidt et al., 2010*; *Ross and Plano, 2011*), as structural information is lacking. In particular, the TM domain of YscD was modeled as an α-helix and equilibrated in a membrane bilayer using MD; it connects the two ring densities of YscD on both sides of the IM without any structural strain. The cytosolic domain of the YscD was modeled on the basis of the X-ray structure of the *Y. pestis* homologue (*Lountos et al., 2012*), and arranged in a regular ring conformation fitting the electron density observed in the cytoplasmic injectisome regions underneath the

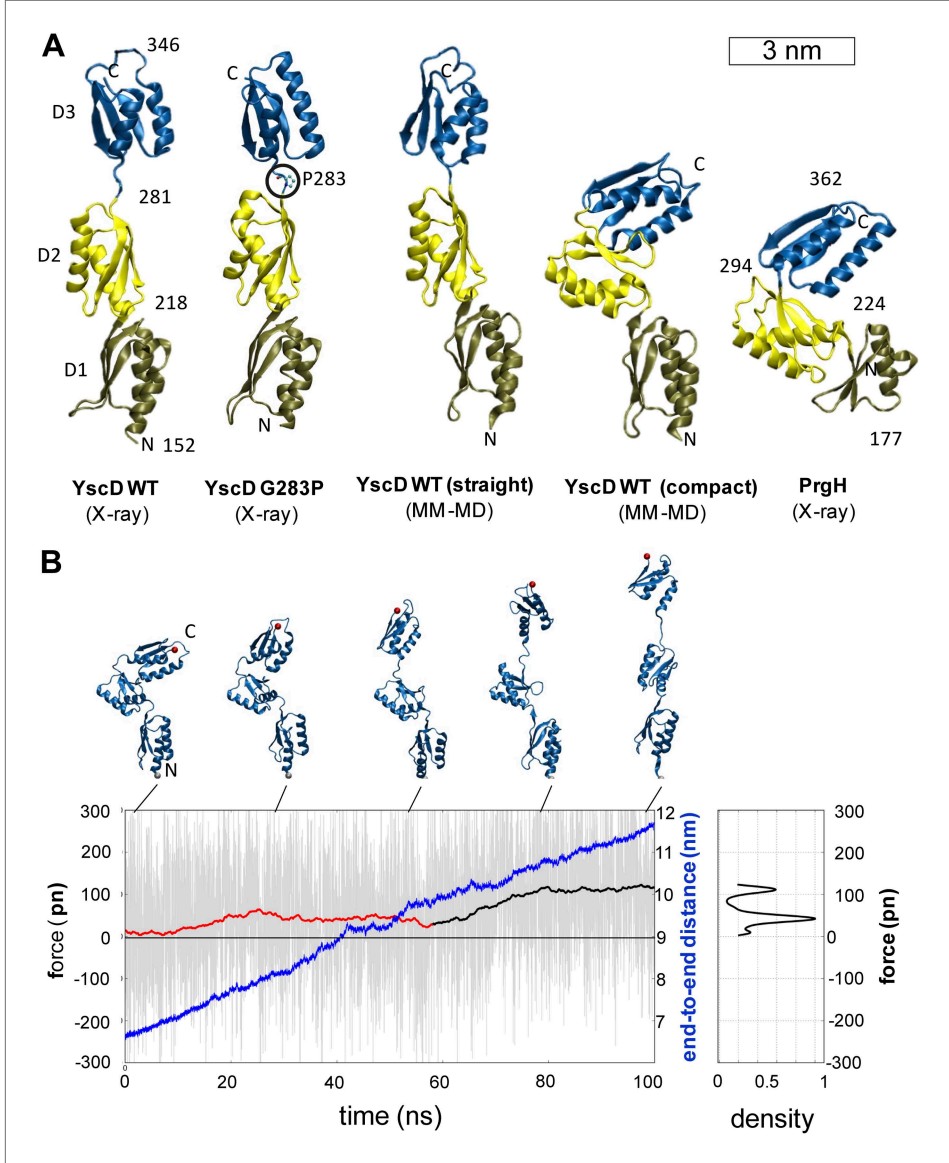

**Figure 3**. Structural elasticity of YscD. (**A**) Comparison of different conformers of YscD with the structure of PrgH (***Spreter et al., 2009***); from left to right: X-ray structures of wild type and G283P mutant (mutation highlighted by a black circle) of YscD, two representative conformations of the elongated and contracted YscD monomer obtained from MD simulations, and X-ray structure of PrgH. (**B**) Force-extension profile from steered MD simulations stretching an YscD monomer; raw data obtained using a pulling velocity v = 0.05 nm/ns are reported (light gray) together with the running average (red and black for extension and unfolding, respectively); end-to-end distance is reported in blue. The inset on the right reports the normalized kernel density profile of the forces: the two distinct peaks correspond, respectively, to the barrier-free extension from the compact to the elongated form, and to the beginning of the unfolding of YscD's third domain.

The following figure supplements are available for figure 3:

**Figure supplement 1**. Comparison of YscD150–362 wt, YscD150–347 G283P, and PrgH crystal structures.

**Figure supplement 2**. Effect of G238P mutation on YscD elasticity.

**Figure supplement 3**. Mutation G283P in YscD has no impact on type III secretion compared to E40 (WT) and in trans complemented YscD.

IM (*Figure 2A*). The cryo-ET map did not allow a direct density fit in the radial direction, due to the limited resolution of the map; the density corresponding to YscJ was smeared. However, the map revealed the height of the periplasmic domains of YscD and provided a hint for the radius or the ring and thus multimerization state of YscDJ. The assembly of the YscDJ ring (*Figure 4*) revealed that only an elongated arrangement of the C-terminal periplasmic domain of YscD—similar to the X-ray structure—best fits the in situ cryo-ET map, extending up towards the position of the secretin ring. Due to the lack of structural information for the last 70 amino acids of YscD, our model did not include this region, which is responsible for connecting to the secretin YscC (*Spreter et al., 2009*; *Schraidt et al., 2010*; *Ross and Plano, 2011*). Most of the YscD/YscJ interactions occur between the first periplasmic domain of YscD and the C-terminal domain of YscJ. The interacting domains are both tightly anchored to the inner membrane by transmembrane helical segments, providing further stability to the YscDJ ring, and leaving the second and the third domains of YscD free to stretch and interact with the YscC secretin ring.

## In situ elongation of the OM secretin YscC and the entire basal body

The structure of an isolated Ysc *Yersinia* injectisome is not available for comparison; therefore we examined *Shigella flexneri* bacteria by cryo-ET and generated an average in situ structure of the *Shigella flexneri* injectisome (*Figure 5*) at ~7 nm resolution. The limiting factor for the resolution was the extreme thickness of the bacteria. Nevertheless, despite the lower resolution, the map allows to reliably detect the positions of the membrane surrounding the injectisome, resulting in a length measurement between the centers of the membranes of 32 nm (*Figure 5C*). This distance measurement was not affected by the resolution of the map (*Figure 5E*).

To evaluate the contribution of the secretin YscC to the elasticity, we expressed and purified full-length YscC, reconstituted it into liposomes, and reconstructed its structure at a resolution of ~3 nm by cryo-ET and sub-volume averaging (*Figure 6*). The isolated and liposome-inserted YscC complexes showed several ring densities: in the membrane bilayer, ~5.5 nm below the membrane, and ~12 nm below the membrane. A comparison with the ring densities in the in situ reconstruction of the entire injectisome lets us tentatively assign the last ring of the isolated YscC to the third ring density 17 nm below the membrane (*Figure 6B*). We speculate that this density corresponds to the YscC–YscD junction and also contains the fourth periplasmic domain of YscD of 70 aa, which might affect the length of the YscD structure by 1–3 nm, depending on the type of molecular interaction between YscC and YscD. A higher-resolution structure of the YscC–YscD interaction would be needed to determine the exact dimensions. The described assignment of the YscC/YscD junction would correspond to a 30–40% length extension of YscC in the assembled injectisome, with respect to its lipid vesicle-reconstituted state (*Figure 6B*).

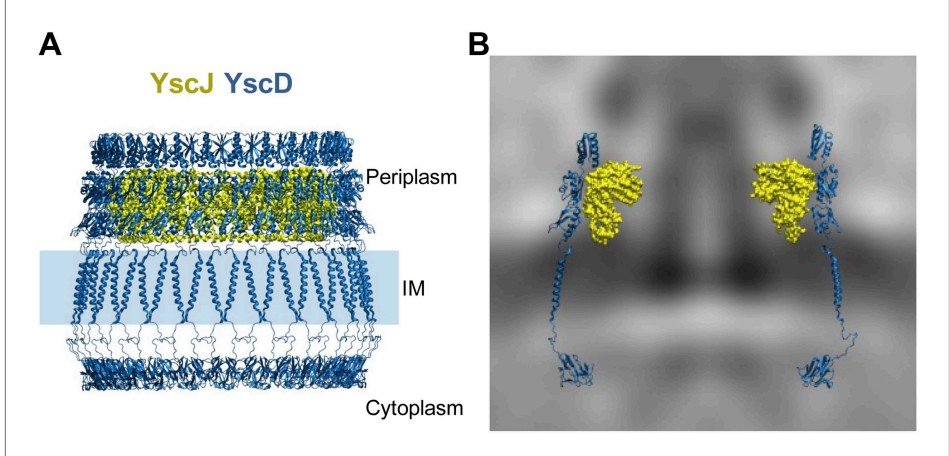

**Figure 4**. Structural assembly of YscDJ ring at the basal body. (**A**) Side view of the generated 24-mer ring model of YscD (blue) and YscJ (yellow). Each YscD subunit has been extended by the transmembrane helix segment and the N-term cytosolic domain. The position of the IM is indicated by a blue area and is positioned according to the IM observed in cryo-ET on the right (**B**). (**B**) Overlay of the injectisome cryo-ET map with the atomistic YscDJ ring model from (**A**). Only two YscDJ units have been superimposed on the cryo-ET map section for clarity. DOI: 10.7554/eLife.00792.012

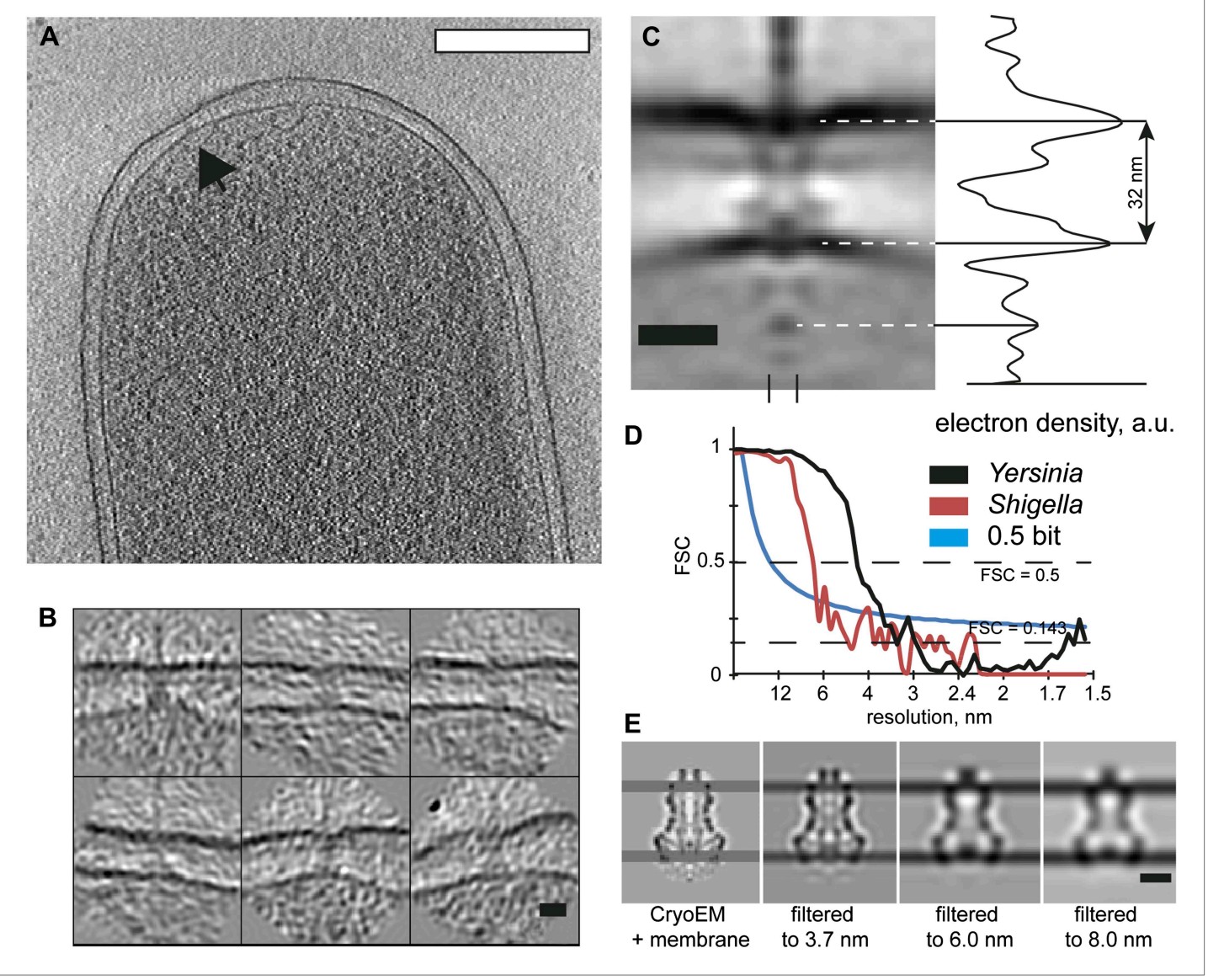

**Figure 5**. Visualization and structure of the *Shigella flexneri* injectisomes in situ. (**A**) 30-nm thick section through a tomogram of an *S. flexneri* cell. Arrow points to the basal body of an injectisome. Scale bar: 300 nm. (**B**) Typical views of *S. flexneri* injectisomes oriented vertically. Scale bar: 20 nm. (**C**) Left: average structure of the *S. flexneri* injectisome in situ. Scale bar: 20 nm; right: electron density along the 8 nm profile indicated with two dashes (bottom). (**D**) Right: comparison of Fourier shell correlation for *S. flexneri* (red) and *Y. enterocolitica* (black); Blue line: 0.5 bit information threshold. Resolution of *S. flexneri* is 7 nm, *Y. enterocolitica* is 4 nm (0.5 criterion). (**E**) Resolution limitation applied to the single particle cryo-EM map (EMD 1871) placed between two added membrane densities. A lower resolution does not affect the visible inter-membrane distance. Scale bar: 10 nm.

## Basal body of the injectisomes elongate under osmotic pressure

In order to influence the intermembrane distance, we placed *Y. enterocolitica* bacteria in high salt media (10-fold concentrated PBS; 10× PBS). The osmotic pressure experienced by the bacterial membranes (**Koch, 1998**) significantly widened the periplasmic space, while injectisomes remained in place and were studied by cryo-ET. The length of observed basal bodies increased by ~10% (with statistical significance, *Figure 7*), while preserving the basal bodies seemingly intact.

## Discussion

Here we present the first visualization of injectisomes from *Yersinia enterocolitica* and *Shigella flexneri* inside intact bacterial cells. Despite an observed length variation among *Yersinia* injectisomes,

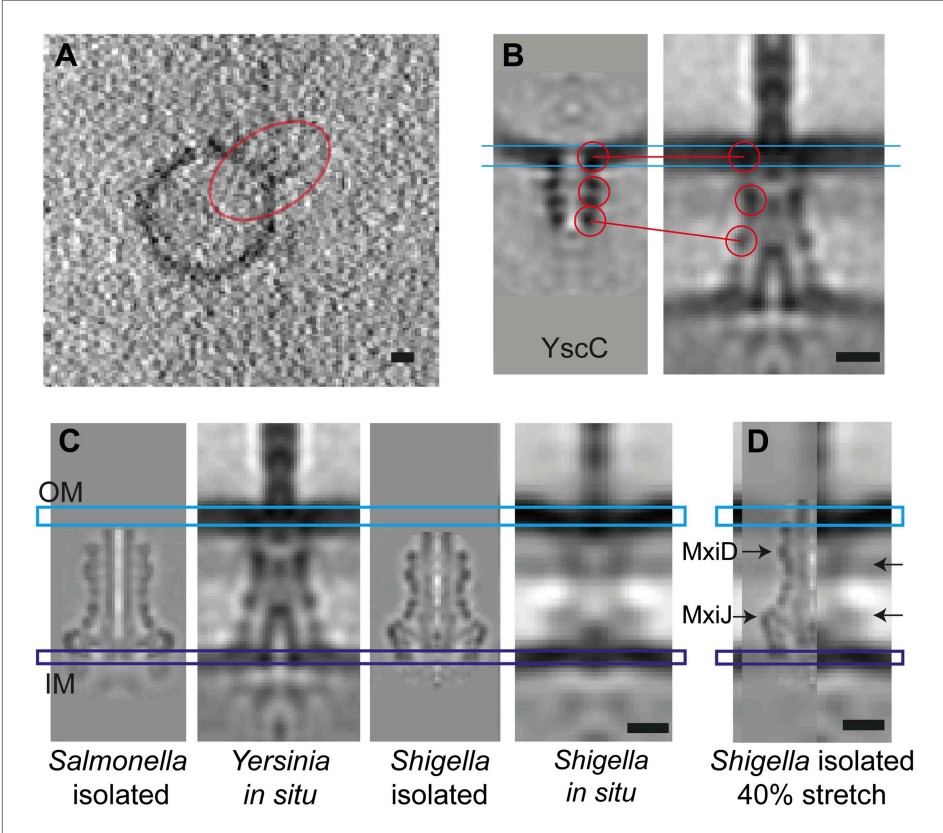

**Figure 6**. Elongation of the in situ structure over the isolated versions. (**A**) 8-nm thick section through a single tomogram of an YscC multimer reconstituted into a lipid vesicle. (**B**) Average structure of liposome-reconstituted YscC (left), and matching densities in the *Y. enterocolitica* injectisome (right). (**C**) Comparison of the *Y. enterocolitica* and *S.flexneri* in situ injectisomes with high-resolution single particle structures of *Salmonella enterica* SPI-1 (EM Data Bank entry EMD 1617), and *Shigella flexneri* (EMD 1871). Blue and purple bars indicate the outer (OM) and inner (IM) membranes. (**D**) The 40% stretched structure of the isolated *S. flexneri* injectisome (right) overlaid onto the *S. flexneri* injectisome in situ (left). Arrows indicate positions of recognizable densities of MxiDJ in the in situ and the stretched-isolated structures. Scale bars: 10 nm.

sub-tomogram averaging allowed reconstructing the needle systems at a resolution of 4 nm, using minicells and applying focal pair tomography (***Kudryashev et al., 2012b***), automated tomogram acquisition, and sub-volume alignment and classification in *Dynamo* (***Castano-Diez et al., 2012***). The in situ reconstructions of the *Yersinia* and *Shigella* injectisomes show high overall similarity considering the lower resolution of the *Shigella* injectisome: the overall dimensions, ring hierarchy and the peptidoglycan attachment location are conserved. We could detect a significant difference in the location of the torus ring assigned to the N-terminus or YscV/MxiA (***Abrusci et al., 2013***). In the *Yersinia* injectisome, it was ~5 nm below the membrane, while in the *Shigella* injectisome it was ~10 nm below the membrane. In the flagellar motor system the distance between the torus and the membrane is also ~10 nm (measured from Figure 5B in ***Abrusci et al. [2013]***, all distances were measured between centers of mass in the EM densities). This difference likely cannot be explained by sequence differences between YscV and MxiA, which are highly homologous (ClustalW similarity score 40). Further experiments or higher resolution structures in situ may give insights into these differences. The distance between the export gate and the ATPases were 11–13 nm for all three assemblies.

Injectisomes contain proteins homologous to FliN and FliM, termed YscQ in *Yersinia* and Spa33 in *Shigella*, which are suggested to form a C-ring (***Morita-Ishihara et al., 2006***) similar to bacterial flagellar motors. However, the in situ reconstructions of the *Yersinia* and *Shigella* injectisomes did not reveal obvious C-rings (***Figure 2A,B*** and ***Figure 3C***). This could be caused by YscQ being

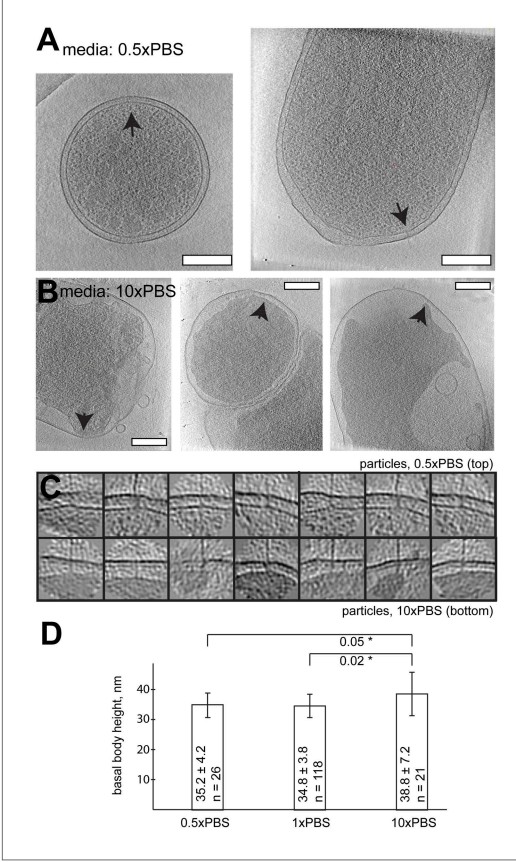

**Figure 7**. Length of basal bodies of injectisomes from *Y. enterocolitica* exposed to media with different osmolarities. (**A**, **B**) Slices though tomograms of bacteria placed into 0.5× PBS diluted with H$_2$0 1:1 (**A**) and into concentrated 10× PBS (**B**). Black arrowheads point to injectisomes. Scale bar in the micrograph: 300 nm. (**C**) 15-nm thick sections though sub-tomograms with roughly vertically aligned particles from 10× PBS media (top) and 0.5× PBS (bottom). Box size: 192 nm. (**D**) Average membrane-to-membrane distances around basal bodies in nanometers with significance tests. Differences between 10× PBS and 0.5× PBS and between 10× PBS and PBS are significant (p=0.05 and p=0.02), while the difference between PBS and 0.5× PBS is not significant (p=0.6).

unstructured, or being only transiently associated with the injectisome. Further investigations are required to address this question.

*Yersinia* injectisomes were found to vary in length by 20% inside the bacterial cells (an inter-membrane distance between 30 and 36 nm). Most injectisomes seemed intact; nevertheless, we cannot exclude that under a certain osmotic pressure the injectisomes may detach from the outer membrane. Biophysical experiments would be required to understand, to which limits the basal body may stretch. In addition to the observed length variations, the injectisome's basal bodies were longer in the native context than suggested from single particle structures: the *Shigella* injectisome in situ appeared about 10 nm or ~40% longer (32 nm vs 22 nm at the membrane level). Dimensions of the *Yersinia* injectisome in situ are similar to those of the *Shigella* injectisome. The elongation of the *Yersinia* injectisome's basal body seems anisotropic with higher contribution from the secretin YscC, based on cryo-ET studies of isolated and membrane-reconstituted YscC. Stretching of YscC by an estimated 5 nm and an elongation of YscD by an estimated 3.5 nm would explain an ~8.5 nm overall extension of the *Yersinia* injectisome's basal body; the remaining 1.5 nm towards the observed 10 nm elongation of the *Shigella* system may be attributed to inter-species differences, or the limited resolution of the maps. High resolution structures are available for the periplasmic domains of PrgH (*Spreter et al., 2009*), YscD (this work), EscJ (*Crepin et al., 2005*; *Yip et al., 2005*) and secretin EscC (*Spreter et al., 2009*). A higher-resolution structure of the membrane domains of the secretins in combination with further molecular simulations may allow a full mechanical understanding of the observed elasticity of the basal body.

We further compared the in situ structure of the *S. flexneri* injectisome with the available cryo-EM reconstructions of the isolated *S. flexneri* (EMD-1617) (*Hodgkinson et al., 2009*) and *Salmonella typhimurium* SPI-1 injectisomes (EMD-1871) (*Schraidt and Marlovits, 2011*) (*Figure 6C*). The lengths and lateral diameters of the in situ injectisomes from *Y. enterocolitica* and *S. flexneri* are similar, while both isolated structures from *S. flexneri* and *Salmonella typhimurium* are much shorter. Further, the isolated *S. flexneri* injectisome has a slightly larger diameter. Even the shortest of our MRA class averages of in situ *Yersinia* injectisomes (*Figure 2B*) is still longer than the isolated *Shigella* injectisome, which would have to be stretched by ~40% to match the dimensions of the average in situ *Yersinia* structure (*Figure 6D*). This is in agreement with the possible 8.5 nm length variation imparted to *Y. enterocolitica* injectisomes by YscC and YscD.

Our combined analysis of the injectisome by cryo-ET and image processing, X-ray crystallography, and MD simulations resulted in a pseudo-atomistic model of the YscDJ ring at the bacterial IM (*Figure 4*). At the currently available resolution, the cryo-EM map doesn't allow any conclusions about the interaction between the YscD and YscJ protomers. However, the map gives insights about the size of the assembly

and the orientation of the protomers with respect to the bacterial IM. In particular, the 24-meric YscD assembly is compatible with the basal body's diameter, as observed by cryo-EM, while a higher number of YscD units would lead to a poorer match with the experimental data. Moreover, the ring model for the TM and cytosolic domain of YscD further confirms a 24-mer stoichiometry as observed from the optimal fit with the cryo-EM density. The construction and docking of the YscJ 24-mer further supports the proposed stoichiometry. Taking the observed stoichiometry of EscJ into account (*Crepin et al., 2005*; *Yip et al., 2005*), the model constructed of 24 YscJ molecules does not produce gaps or overlaps in the final structural assembly, as we otherwise obtained when using a higher or lower stoichiometry. Moreover, while we cannot rule out a different ring arrangement for YscJ in *Yersinia* compared to EscJ in *E.coli*, the outer diameter of the assembled YscJ 24-meric ring matched well the inner diameter of a YscD 24-meric ring.

Modeling the TM helix segment connected to the first periplasmic domain (residues 152–218) gave additional hints on the YscD orientation at the membrane surface. The short loop linking the two domains does not allow for large flexibility, providing a solid anchor point for the first YscD periplasmic domain at the membrane surface, and forcing a preferential perpendicular orientation to the IM. MD simulations also suggested a limited fluctuation for this portion of YscD embedded in a membrane model, confirming the preferential orientation found in the 24-mer YscD ring assembly (*Figure 4*).

YscDJ ring modeling has been performed to test for any elasticity of the basal body. Several models were constructed based on the extended and compact forms of the YscD monomer explored by steering MD (as described above, 'Materials and methods'). Although the cryo-ET density could be optimally fitted only by extended forms similar to the X-ray YscD conformation, MD snapshots of individual monomers in more compact forms (*Figure 3—figure supplement 2*) allowed the construction, without incurring in steric clashes, of YscD 24-mer ring structures representative of possible states of intermediate compression compared to the fully extended ring model.

Bacteria are likely exposed to considerable physical forces in vivo, originating from swimming motions, contact with host cells, or from changes in osmolarity of the surrounding medium. For survival, the bacterial membranes must withstand distance variations, while membrane proteins and complexes must continue to function. We hypothesize that the here observed elasticity might allow the injectisome to better cope with such membrane stresses. This is supported by our finding that exposure of the *Y. enterocolitica* bacteria to osmotic stress from high salt conditions resulted in a significant elongation of their basal bodies (*Figure 7*). It is tempting to speculate on any additional functional role of the observed elasticity, for example, to facilitate substrate secretion through a ratchet pump mechanism (*Astumian, 2011*). Moreover it would be interesting to quantitatively assess the injectisome's basal body elongation capabilities with biophysical techniques, for example, atomic force microscopy and force spectroscopy. Recently, it was suggested that injectisomes may rotate upon action (*Ohgita et al., 2013*), similarly to the evolutionary related flagellar motors. To our knowledge, however, the significantly larger basal bodies of flagellar motors do not vary in length, despite their active rotation. Furthermore, the basal bodies of the bacterial flagellar motors do not seem to differ in dimensions between their isolated and in situ structures (*Chen et al., 2011*).

Recent structural studies in situ revealed flexibility of large molecular complexes. The bacterial type VI secretion system is reported have two long lived stated in the action cycle (*Basler et al., 2012*); bacterial flagellar motors of *Borrelia burgdoferii* partially lack the C-ring (*Kudryashev et al., 2010*) and the stator rings follow the curvature of the cytoplasmic membrane (*Liu et al., 2009*). Protomers of nuclear pore complexes deviate from eightfold symmetry by up to 20% of the diameter (*Beck et al., 2007*), which may be responsive to the turgor pressure of the nuclear envelope (*Akey, 1995*). We here hypothesize that for more fragile protein complexes that span two membranes, tertiary structure elasticity may constitute a general mechanism for structural protection.

## Materials and methods

### Bacterial strains, plasmids, and genetic construction

*E. coli* BW19610 (*Howitt et al., 2006*) used for cloning and *E. coli* Sm10 λ *pir*+ used for conjugation were routinely grown in Luria broth (LB) or on LB agar (LA) plates at 37°C. Streptomycin was used at a concentration of 100 µg/ml to select for suicide vectors. All *Y. enterocolitica* strains are derivates of E40 (*Sory et al., 1995*), where for biosafety reasons six effector genes were deleted, as well as the *asd*

gene. They were routinely grown at 25°C in brain heart infusion (BHI) broth containing 35 µg/ml nalidixic acid. To allow growth of *asd* mutant strains, the medium was supplemented with 50 µg/ml meso-diaminopimelic acid. *Shigella flexneri* SC560 (*Sansonetti, 1991*) were routinely grown at 37°C in BHI containing 100 µg/ml streptomycin. Mutator plasmid pMK3 was made by amplification of the *asd* 5′ region with oligos 3541/3543 and the 3′ region with oligos 3542/3544. The 5′ region was digested with *Sal*I/*Eco*RI and the 3′ region with *Eco*RI/*Xba*I. Both fragments together were ligated into the *Sal*I/*Xba*I restriction site of pKNG101. To construct pMA87, flanking regions of about 250 bp just upstream and downstream of *minD* were amplified from purified genomic DNA from *Y. enterocolitica* E40 using oligonucleotides 6416/6417 and 6418/6419 respectively (*Supplementary file 1C,D*). The two fragments were joined by overlapping polymerase chain reaction (PCR), and the resulting fragment was cloned into the *Sal*I/*Xba*I restriction sites of suicide vector pKNG101 (*Kaniga et al., 1991*). To construct pMA6, full-length *yscC* with a stop codon was amplified from the pYVe40 plasmid using primers 5013/5014 and introduced into the *Nco*I/*Eco*RI restriction sites of pBAD/mycHisA.

Cultures were inoculated at an optical density ($OD_{600}$) of 0.1 in BHI broth containing sodium oxalate (20 mM) (BHI-OX) supplemented with glycerol (4 mg/ml) and $MgCl_2$ (20 mM). After 2 hr of growth at 25°C, induction of the *yop* regulon was performed by shifting the culture to 37°C (*Cornelis et al., 1987*). Expression of the pBAD constructs was induced by adding 0.03% L-arabinose to the culture just before the shift to 37°C. After 4 hr of incubation at 37°C, cultures were used for further analysis.

## *Y. enterocolitica* minicells generation

Mutant strains forming minicells were generated by a two-step allelic exchange (*Kaniga et al., 1991*). The *Y. enterocolitica* parent was mated on a plate with *E. coli* Sm10 λ pir⁺ containing the corresponding mutator plasmid. To select for integration of the mutator plasmid the conjugation mix was plated on nalidixic acid and streptomycin. In a second step, the streptomycin selection pressure was released during several generation times allowing the excision of the mutator plasmid. Plating on LB agar containing 5% sucrose allowed selection for colonies that underwent the second recombination step and had lost the mutator plasmid. These colonies were screened for the mutant allele by colony PCR. As an exception, yadA mutants were made by insertion of the entire mutator plasmid pLJM31 into yadA. To avoid wild-type revertants by excision of the plasmid, constant streptomycin selection was applied.

## Secretion analysis and immunoblotting

Bacteria and supernatant (SN) fractions were separated by centrifugation at 20,800 g for 10 min at 4°C. The cell pellet was taken as total cell (TC) fraction. Proteins in the supernatant were precipitated with trichloroacetic acid 10% (wt/vol) final for 1 hr at 4°C. SN and TC fractions were separated on a 12% or 15% SDS-PAGE, respectively. In each case, proteins secreted (SN) by $3 \times 10^8$ bacteria or produced (TC) by $1 \times 10^8$ bacteria were loaded per lane. Immunoblotting was carried out using rabbit polyclonal antibody against YscD (internal number MIPA232; 1:1000). Detection was performed with the swine anti-rabbit secondary antibodies conjugated to horseradish peroxidase (1:5000; Dako), before development with the LumiGLO Reserve chemiluminescent substrate (KPL).

## Fluorescence microscopy

For fluorescence imaging, about 2 µl of bacterial culture (see above) were placed on a microscope slide layered with a pad of 2% agarose in PBS. A Deltavision Spectris optical sectioning microscope (Applied Precision, Issaquah, WA) equipped with an UPlanSApo 100×/1.40 oil objective (Olympus, Tokyo, Japan) and a coolSNAP HQ CCD camera (Photometrics, Tucson, AZ) was used to take differential interference contrast (DIC) and fluorescence photomicrographs. GFP filter sets (Ex 490/20 nm, Em 525/30 nm) were used for GPF visualization. DIC frames were taken with 0.1 s and fluorescence frames with 1.0 s exposure time. Per image, a Z-stack containing 20 frames per wavelength with a spacing of 150 nm was acquired. The stacks were deconvolved using softWoRx v3.3.6 with standard settings (Applied Precision). A representative DIC frame and the corresponding fluorescence frame were selected and further processed with the ImageJ software.

## YscC purification and reconstitution on liposomes

The pYV–cured *Y. enterocolitica* strain carrying plasmids pMA6 and pRS6 (*Allaoui et al., 1995*) containing the *yscC* and *yscW* genes, respectively, was grown in BHI broth. To induce expression of

*yscC*, bacteria were inoculated at $OD_{600}$ = 0.1 in BHI-Ox broth supplemented with glycerol (4 mg/ml), $MgCl_2$ (20 mM), ampicillin (100 mg/ml), nalidixic acid (25 μg/ml), and tetracycline (10 μg/ml). The culture was grown for 2 hr at room temperature, induced with 0.05% arabinose and grown for 6 hr at 37°C. The entire YscC purification was performed on ice. Bacterial cells were washed with 0.9% NaCl, resuspended in 50 mM Tris-HCl pH 8.5 and 1 mM EDTA and disrupted using a sonicator. The membrane fraction was isolated by ultracentrifugation for 1 hr at 150,000×*g* (4°C), and membrane proteins were solubilized in buffer containing 2% DDM (n-dodecyl-β-D-maltopyranoside, Anatrace), 50 mM Tris-HCl pH 7.8, 250 mM NaCl, 5 mM EDTA and protease inhibitor (complete protease inhibitor, Roche) for 1.5 hr at room temperature. Insoluble material was removed by ultracentrifugation for 1 hr at 150,000×*g* (4°C). After the addition of sucrose to a final concentration of 15% (wt/wt), the extracted membrane proteins were layered on top of a 20–40% (wt/wt) sucrose gradient in gradient buffer (0.04% DDM, 50 mM Tris-HCl pH 7.8, 250 mM NaCl, 5 mM EDTA, protease inhibitor) and centrifuged at 38,000 rpm in an SW41 rotor (Beckman) for 30 hr. Fractions containing YscC were dialyzed against chromatography buffer (0.04% DDM, 10 mM Tris-HCl pH 7.8, 100 mM NaCl, 0.1 mM EDTA) and loaded on MonoQ 5/50 GL IEC (GE Healthcare). YscC was eluted at 400–500 mM NaCl. The pure YscC oligomers were separated from YscC oligomer dimers and small contaminants by gel filtration using a Superose 6 10/300 GL column (GE Healthcare). Fractions containing YscC were stored at −20°C for electron microscopy.

To reconstitute YscC into liposomes, the purified secretin (0.2 mg/ml) was mixed with DDM-solubilized *E. coli* polar lipids at 5:1 lipid-to-protein ratio and vigorously mixed overnight with Bio-Beads (Bio-Rad, Hercules, CA) at room temperature. Cryo-EM imaging (see below) showed that most of the proteins were facing outside of the lipid vesicles.

## Sample processing for CEMOVIS

Samples we prepared according to the protocol described elsewhere, with slight modifications (*Bleck et al., 2010*). In brief, for cryo-electron microscopy of vitrified sections (CEMOVIS), gently spun bacteria were resuspended in a final concentration of 20% dextran in phosphate buffer system (PBS, dextran: average molecular mass 40 kDa; Sigma-Aldrich). Afterwards the mixture was introduced into specimen copper tubes (Cat.# 16706871, Leica Vienna, Austria) and vitrified with an EMPACT-2 high-pressure freezer (Leica). Ultrathin sections (50–60 nm) from vitreous cells were obtained using a FC7/UC7-ultramicrotome (Leica). Sections were collected on Quantifoil grids (3.5/1), and mounted into Titan Autoloader cartridges (FEI, Eindhoven, Netherlands). Imaging was done with an FEI Titan Krios (300 kV accelerating voltage, Cs = 2.8 mm) at a nominal defocus of −6 μm; images were recorded on a Gatan US4000 CCD camera; the total electron dose for imaging was kept below 4000 e/nm$^2$.

CEMOVIS yielded an average inter-membrane distance of 35 nm measured between the centres of electron density (*Figure 2—figure supplement 2*), documenting that the plunge-frozen bacteria for the injectisome structure studies had not suffered significant dehydration or changes in buffer osmolarity prior to rapid freezing. *E. coli* K12 minicells have an average membrane to membrane distance of slightly over 30 nm (*Liu et al., 2011*), and the evolutionary related bacterial flagellar motors show inter-membrane distances in the range of 28–40 nm (measured from Figure 1 in reference *Chen et al. [2011]*).

## Cryo electron tomography

*Yersinia enterocolitica* cells or minicells or *S. flexneri* SC560 were supplemented with 5% of 10 nm gold beads, placed on holey carbon grids (Quantifoil Micro Tools GMBH, Germany), quickly vitrified using a FEI Vitrobot IV (FEI Corp, Hillsboro), and imaged at liquid nitrogen temperatures in an FEI Titan Krios (FEI Corp, Hillsboro) operated at 300 kV acceleration voltage and equipped with a GIF and an US1000 CCD camera (Gatan Inc, Pleasanton). The magnification calibration of the FEI Titan Krios and GIF/CCD detectors was verified, using a gold lattice, vitrified tabacco mosaic virus sample, and graphene (*Pantelic et al., 2011*), and found to be precise to better than 2%. Tomograms were collected at 2 or 3° increments over a 120° range. The total electron dose was less than 10,000 electrons/nm$^2$ for regular tomograms and less than 20k electrons/nm$^2$ for focal pair tomograms (*Kudryashev et al., 2012b*). Tomograms were aligned with the aid of the gold beads using the *eTomo* software (*Kremer et al., 1996*), and reconstructed using Matlab based Dynamo scripts (*Castano-Diez et al., 2012*).

## Image processing procedures for sub-volume averaging

High defocus tomographs from focal pairs and tomograms of minicells were aligned by gold marker fiducials using *eTomo* (*Kremer et al., 1996*) and reconstructed by weighted back projection using custom written Matlab scripts. Central positions and directions of needles were manually determined

for 421 injectisomes from the tomograms of minicells and for 1490 injectisomes from tomograms of regular sized cells acquired as focal pairs. Injectisomes were extracted to volumes of 128×128×128 voxels. Low-defocus (high resolution) particles were generated by a combination of global high- and low-defocus tilt series alignment, followed by refinement of patches of micrographs around the injectisomes. This used the 'local feature refinement' method described in more details in reference (*Kudryashev et al., 2012b*). From 1490 particles, 520 were selected for high-resolution processing based on having good correlation of high- and low-defocus injectisome volumes to each other. In addition, some tomograms of particles that did not contain projections of the injectisomes in all low-defocus tilt series images were also discarded.

An initial average structure was produced as a sum of all injectisomes with the volumes rotated such that the needles were pointing into the same (vertical) direction. Next, multiple rounds of alignment with restricted angular rotation ranges were performed on high-defocus particles and on minicell dataset particles, considering only voxels within a mask on the needle and the outer membrane area with a pixel size of 1.48 nm (*Figure 2—figure supplement 1*). The information about the missing wedge was used to constrain correlation during alignment of particles to the average, and appropriate Fourier component weighting was performed during generating the average at the end of each iteration. Next, two independent alignments were calculated with two different soft masks: one containing the outer membrane and the needle structures, and another one containing the cytoplasmic membrane and the needle structure (*Figure 2—figure supplement 1*). During alignment we imposed 19-fold axial symmetry to the reference at the start of each iteration, while we applied 12-fold rotational symmetry to the final structures. The two resulting structures were aligned against each other by cross correlation maximization. Cropping them together approximately in mid-height between the two membranes produced a merged structure. The resulting volume was limited to 4 nm resolution, which was determined from gold standard Fourier Shell Correlation (FSC) processing using the FSC = 0.143 offset (*Figure 2—figure supplement 1D*, and 'Gold standard FSC image processing'). The FSC in *Figure 2—figure supplement 1C* was produced as an average FSC inside the two used alignment masks. The processing was done by AV3 processing package for Matlab (*Forster et al., 2005*), in-house written scripts, and our Dynamo software tool for user-friendly sub-tomogram averaging (http://www.dynamo-em.org) (*Castano-Diez et al., 2012*).

For the initial alignment of YscC we manually clicked into the membrane part and inside the liposome in order to establish the initial orientation of the molecule for 282 particles. We used a featureless plane with a ball as an initial reference for the alignment, after which the half of particles with higher correlation coefficient contributed to the reference for next iteration.

Volume-rendered visualizations were produced semi-automatically with Amira (http://www.amira.com). The reconstruction of the *Y. enterocolitica* injectisome will be deposited to the EMD upon acceptance of the manuscript.

## 'Gold standard FSC' image processing

In order to validate the resolution of the averaged injectisome structure, we randomly separated 624 manually pre-aligned particles into two independent sets and processed them independently with the same parameters (*Scheres and Chen, 2012*) using Dynamo, performing the following steps:

1. Two initial reference structures were generated using the parameters of the initial manual pre-alignment. These structures were noisy and unstructured.
2. Restricted iterative alignment of each set independently was performed, using an alignment mask on the basal bodies and the needle. A rough alignment and averaging was performed with particles from the highly defocused dataset. In subsequent iterative alignments, these particles were replaced by the 'low defocus' particles. During the alignments, a low pass filter at 6 nm resolution was used, and a high rotational symmetry was applied.

The final independent structures for the two sets were compared by Fourier shell correlation (*Figure 2—figure supplement 1D*), indicating a resolution of ~4 nm by the 'gold standard' criterion (FSC = 0.143).

## MRA classification of sub-volumes

Iterative multi reference alignment (MRA) was performed on injectisome sub-volumes within wide elliptical, Gaussian smoothed mask, areas with an extra weight on the needle area. 10 initial references were produces from the average structure with an addition of low, 10% Gaussian noise. Further,

starting from the alignment that produced the average structure, each of the selected particles was aligned to each reference and finally contributed only to the reference to which it had the highest correlation coefficient. The maximum angular increment allowed was 4°; the maximum shift was 2 voxels. While our reconstruction made from all injectisomes did not reveal the outer second periplasmic density layer, our MRA data showed in the majority of sub-volume class averages that outer second periplasmic layer at different positions, suggesting that its height also varies among the individual injectisomes with respect to the cytoplasmic membrane. The inner (bottom) periplasmic layer was less well visible in the class averages, suggesting a less defined contact between it and the basal body. We also tried PCA+K-means classification however it contained alignment bias while MRA was free from it due to iterative alignment.

## YscD production and purification for crystallization

Cultures were launched from *E. coli* Tuner (DE3) (transformed with pUWSS2 or with pUWSS3) in LB/Amp overnight at 37°C. Cells were diluted to an $OD_{600}$ of 0.1 in 2×1L LB media with ampicillin (final concentration 100 µg/ml) at 37°C. Protein production was started by adding of 0.2 mM isopropyl-β-D-thiogalactopyranoside (IPTG) at an $OD_{600}$ of 0.6–0.8. To avoid inclusion bodies temperature was lowered to 20°C and cells were further incubated for up to 18 hr. Cells were harvested by centrifugation at 6000×*g*, 4°C for 15 min, and resuspended in 1× PBS. Cell lysis was carried out either by cell disrupter (Constant Systems Ltd, Kennesaw, GA) or by sonication. Cell debris was separated from protein solution by centrifugation at 16,000 rpm (rotor SS-34) for 40 min. GST-YscD variants were batch bound on Protino Glutathione Agarose 4B beats (Macherey and Nagel), which had been equilibrated in 1× PBS. Unbound protein was washed with 12 column volumes (CV) of 1× PBS and with 8 CV of protease buffer (50 mM Tris, pH 7.5, 150 mM NaCl, 1 mM DTT, 1 mM EDTA). Loaded beats were resuspended in 10 ml of protease buffer. YscD variants were cleaved from the GST-tag by addition of 200 units of PreScission Protease (GE Healthcare) and kept overnight at 4°C. The YscD[150–362] or YscD[150–347] G283P protein in the supernatant was used for further purification steps. YscD[150–362] was dialyzed in ion exchange column (IEC) buffer A (20 mM HEPES pH 7.0, 60 mM NaCl at 4°C) and impurities bound on a MonoQ 10/10 column (GE Healthcare). Flow through contained YscD[150–362], which was concentrated and finally polished via gel filtration (Superdex 75 16/60; GE Healthcare) in IEC buffer A. In contrast YscD[150–347] G283P supernatant was directly concentrated and one step purified on a Superdex 75 26/60 size exclusion column (GE Healthcare) using the same buffer conditions as for YscD[150–362]. Pure protein fractions were pooled concentrated to 3–6 mg/ml, flash frozen in liquid nitrogen and stored at −80°C. The identity and integrity of YscD variants was confirmed by N-terminal sequencing and mass spectrometry (HZI-in house).

## Crystallization, data collection, and model building of YscD[150–362]

YscD[150–362] crystals for micro-seeding were obtained by mixing equal volumes of YscD[150–362] (3–6 mg/ml in IEC buffer A) with precipitant solution (0.2 M NaH$_2$PO$_4$, 25% [wt/vol] PEG 3350) in hanging-drop vapor-diffusion crystallization trays. Crystal clusters grew in 2–3 days at 20°C. Thereafter micro-seeding techniques were applied to grow large and single crystals in 0.2 M NaH$_2$PO$_4$, 11–13% (wt/vol) PEG 3350 with a protein concentration of 3–6 mg/ml at 20°C. Prior to data collection crystals were stepwise cryo-protected in 20% (wt/vol) PEG 3350, 0.2 M NaH$_2$PO$_4$, 15% (vol/vol) glycerol. Native data were collected at 100 K at the 'Deutsches Elektronen-Synchrotron' (DESY, beamline X11 in Hamburg). Iodine SAD phasing was performed after a published protocol (*Dauter et al., 2000*) using the Cu K$_α$ radiation of a Rigaku MicroMax 7HF Cu anode equipped with a Saturn 944+ detector. Therefore YscD[150–362] crystals were soaked for 30–60 s in 500 mM KI, 40% (vol/vol) glycerol, 15% (wt/vol) PEG 3350, 0.2 M NaH$_2$PO$_4$ and immediately flash frozen. Data sets were indexed, integrated, and scaled with the XDS/XSCALE package (*Kabsch, 2010a, 2010b*). The anomalous signal of iodine (d″/σ > 1.3) was used to 2.6 Å to solve the structure with the SAS and MRSAD protocol of Auto-Rickshaw (*Panjikar et al., 2009*). An initial model from amino acids 152–347 was built by ARP/wARP (*Morris et al., 2003*) and manually inspected and rebuilt using COOT (*Murshudov et al., 1997*). Refinement was carried out with *Refmac5* (*Murshudov et al., 1997*) from the CCP4 suite (*Collaborative Computational Project, 1994*). This model was used as a search model for the native dataset of YscD[150–362], which also poorly refined to 2.7 Å and hence was not deposited at the Protein Data Bank (http://www.pdb.org).

## Crystallization, data collection, and model building of YscD$^{150–347}$ G283P

YscD$^{150–347}$ G283P crystals grew from equal volumes of protein (5.8 mg/ml in IEC buffer A) with precipitant solution (0.15 M NaH$_2$PO$_4$, 20% [wt/vol] PEG 3350, 60 mM NaCl) in hanging-drop vapor-diffusion crystallization trays at 20°C (EasyXtal; QIAGEN). Crystals appeared after several days and reached full size (360 × 270 μm) after 2–3 weeks. Similar to the wild-type crystals YscD$^{150–347}$ G283P crystals were sensitive for any tested cryo-protection. Hence crystals were flash-frozen in crystallization condition without any cryo-protection and a dataset collected at 100 K at the BESSY (Berlin, MX-14.1). Crystals of G283P YscD$^{150–347}$ diffracted to 1.4 Å in the same space group as YscD$^{150–362}$, P2$_1$, but with different cell dimensions (YscD$^{150–362}$: a = 48.2 Å, b = 29.8 Å, c = 69.9 Å, α = γ = 90°, β = 97.1°; YscD$^{150–347}$ G283P: a = 38.1 Å, b = 51.7 Å, c = 50.8 Å, α = γ = 90°, β = 106°). The data set was indexed, integrated, and scaled with the XDS/XSCALE package (*Kabsch, 2010a*, *2010b*). Phases were obtained with *Phaser* (*Mccoy et al., 2007*), using amino acid range 152–280 from the YscD$^{150–362}$ model as search coordinates. Residues 281–347 were built manually using *COOT* (*Murshudov et al., 1997*) and refined with *Refmac5* (*Murshudov et al., 1997*) from the CCP4 suite (*Collaborative Computational Project, 1994*). The final model of YscD$^{150–347}$ G283P was deposited at the Protein Data Bank (http://www.pdb.org; PDB code: 4alz). Data collection and refinement statistics are displayed in *Supplementary file 1A*.

## Rational design of an YscD mutant to achieve better resolution in X-ray crystallography

The initial crystal obtained from YscD$^{150–362}$ showed a resolution of 2.7 Å (*Figure 3A*). This preliminary structure, as well as sequence-based secondary structure and disorder prediction confirmed the presence of three compact α/β domains separated by flexible linkers. We correlated structure and flexibility in the YscD$^{150–362}$ by performing MD simulations of the preliminary structure (in explicit solvent). The system was subjected to geometry optimization and molecular dynamics; after 10 ns equilibration (of RMSD, density, volume) analysis was performed on the last 70 ns of simulation. The secondary structure elements in each of the three domains of YscD were conserved during the simulation, thus confirming the structural stability of the αββαβ-ring building motif. By comparing the computed atomic positional fluctuation to the experimental β-factor of the wt crystal structure we identified the motility of the third periplasmic domain as a possible cause of the poor quality of the crystal.

To improve the crystal we devised a strategy to restrain the motion of the third domain without affecting the protein fold. Essential dynamics analysis (EDA) was performed on the MD trajectory and anisotropic network model (ANM)—normal mode analysis (NMA) ('Materials and methods' below) was applied to the structure and to representative snapshots of the MD simulation. By analyzing the first modes, as independently obtained by EDA and ANM-NMA, we observed that the most relevant collective motions involved bending and rocking of the third domain with respect to the first two. Moreover the most relevant modes suggested the presence of a hinge between the second and third domain, responsible of most of the protein flexibility and causing large displacements of the third domain (*Figure 3A*, *wt*). To assess the role of this hinge in the observed flexibility we performed a set of in silico mutations aimed at reducing the conformational freedom in YscD. We constructed a single (G283P) mutant by atomic replacement from the preliminary crystal structure and subjected the systems to the same simulation protocol and analysis as used for the wild-type protein (*Figure 3—figure supplement 3*). The proline substitution produced a significant damping in the protein motion and a decrease in the atomic positional fluctuation of the third domain. On the basis of these results we constructed yscD$^{150–347}$ G283P. The purified mutant protein produced more regular crystals displaying an improved diffraction pattern, reaching 1.4 Å resolution, with respect to the wild-type YscD (*Figure 3—figure supplement 1*).

## Elasticity of YscD revealed by molecular simulations

The three domains are, in YscD, arranged along a straight line, while the homologous PrgH protein adopts a more compact boot-shaped arrangement. The high flexibility of YscD, as observed in molecular dynamics simulation and predicted by ANM-NMA calculations, led us to infer that the system can access both straight and bent conformations through a stretching process involving bending/rotation of the tree domains. The YscD models presented here and the available structures of PrgH may represent two possible states, selected by experimental conditions and crystal packing among a large ensemble of accessible conformations.

To test the capability of YscD to access both bent and straight conformational states, we performed free and biased molecular dynamics simulations. We reconstructed the collapsed conformation based on the structure of PrgH (PDB: 3GRO; UniProtKB: P41783) by performing independent structural alignment between each domain (loops excluded) of YscD and PrgH; then we used MODELLER (*Fiser et al., 2003*) to splice together the three YscD fragments and to add and relax the loops (*Fiser et al., 2000*). The collapsed YscD was embedded in a box of water molecules and subjected to geometry optimization and molecular dynamics; after 10 ns equilibration (RMSD, density, volume), analysis was performed on a 70 ns equilibrium simulation. As a model of the straight YscD conformation we used the system constructed from the crystal structures (see above). Additionally, we tested with a similar setup and protocol the compact and straight (as in the X-ray structure) conformation of the G283P YscD mutant. The simulations showed that both straight and bent structures constitute stable conformational states for the YscD protein. As discussed above, the straight conformer showed little variation in the domain arrangement with respect to the initial X-ray structure of YscD, although oscillations of the third domain were observed throughout the simulation. The collapsed conformation diverged slightly from the PrgH structural template, mainly due to a different extension of the flexible loops, and different conformation of the first domain, causing a different domain arrangement; in particular the angle between domains one and two changes from 82° (initial geometry/PrgH) to 24 ± 6° (average during simulation) (*Figure 3A* of the main text). Nonetheless, YscD showed the ability to explore a much compact arrangement, very close to the structure of PrgH.

## Tertiary structure elasticity explains a facile inter-conversion between extended and compact YscD states

Tertiary structure elasticity, defined as the rearrangement of tertiary structure in response to mechanical force, represents the first mode of elastic response to external stimuli. To address the tertiary structure elasticity of the periplasmic domain of YscD, a model of the wild-type YscD$^{152–346}$ protein was subjected to unbiased molecular dynamics (MD) and then to steered molecular dynamics (SMD) simulations. The construction and equilibration of the initial compact conformation is reported above. Starting form the final geometry of the MD simulation (70 ns) the system was extended according to a standard SMD procedure. The Cα of Asp152 was kept fixed to its initial position while the Cα of Ans346 was slowly pulled at the constant velocity of 0.5 Å ns$^{-1}$ to reduce the effects of hydrodynamic drag force (*Hsin and Schulten, 2011*). A constant stretching force of 5 kcal mol$^{-1}$, resulting in a thermal noise deviation of 0.35 Å, was employed to pull the Cα of Ans346 along a fixed direction. The end-to-end distance was thus increased from 6.5 nm to 11.5 nm in 100 ns, with the system opposing a force of 35 ± 15 pN for the first 60 ns (100 nm extension); this value is comparable with the force computed, under similar conditions, for the protein titin (*Hsin and Schulten, 2011*). Further stretching causes the rupture of an increasing number of hydrogen bonds within the β strands of the first and third domain, resulting in a progressive growth of the opposing force (above 150 pN). During the SMD simulation the system undergoes a complete stretching from the compact PrgH-like conformation to the fully extended conformation observed in the YscD X-ray structure. The most prominent degrees of freedom, during the extension process, are the two bending angles between neighboring domains pairs.

Due to its high tertiary structure flexibility, YscD can undergo a stretching of about 5 nm without opposing significant forces. Extension of the YscD periplasmic segment has, thus, a non-negligible impact on the overall height of the basal-body. On the basis of these findings we suppose the YscD flexibility confers to the basal body the capability of reacting to variations on the periplasmic space.

## Molecular assembly of the YscDJ portion of the basal body

A model for a 24-mer YscJ assembly (periplasmic domains) was derived by homology modeling based on the X-ray structure of the *E. coli* EscC homologous (PDB id: 1YJ7) (*Yip et al., 2005*) after generating the assembly using a P6 symmetry group. The YscJ 24-mer features a compact arrangement of circular shape. The surface in contact with the outer leaflet of the inner membrane exposes charged and polar residues suitable for interaction with the lipid phosphate head-groups.

Since no suitable structural template exists for the YscD ring, a particle swarm optimization procedure (*Degiacomi and Dal Peraro, 2013*) was used to create an atomistic model of the whole inner membrane ring, as composed by a 24-mer YscD ring encircling the modeled 24-mer YscJ complex. The optimization was guided by spatial restraints extracted from the cryo-EM maps and crosslinking-derived distances from the *Salmonella* and *Shigella* D and J orthologues (*Sanowar et al., 2010*). Namely, loose restraints

based on the cryo-ET maps (height = 10 ± 2 nm, width 26 ± 2 nm, inner radius = 8 ± 2 nm) were imposed to ensure the height and width of the assembly to be smaller than, respectively, 10 nm (maximum extension of YscD prior unfolding) and 24 nm (the PrgH rings features a width of about 27 nm). 24 monomers were then distributed according to a circular symmetry imposing a radius of 8 ± 2 nm. Other loose restraints were derived from the cross-linking experiments available for the PrgH/PrgK system after sequence alignment with YscD/YscJ (*Sanowar et al., 2010*; *Schraidt et al., 2010*), residues between 153 and 160 in YscD were forced to face the YscJ ring and, in particular the region surrounding YscJ-S214. Conformational states extracted from the simulations of the wild-type YscD monomer were used to take into account the native flexibility of YscD and eventually to build the final assembly. The protocol generated six models, all satisfying the initial restraints and producing similar YscD ring arrangement. One of those was selected and further refined by minimization to produce a final structural model of the YscD 24-mer ring interacting with the YscJ ring previously modeled.

We completed the periplasmic part of the YscD ring with the transmembrane (TM) domain and the small globular cytoplasmic domain. The YscD N-terminal cytoplasmic domain has been recently crystallized for the *Y. pestis* homologue (*Lountos et al., 2012*), and was modeled using this template, whereas the helical structure of the TM region and its orientation with respect to the membrane bilayer were assessed through MD simulations. The ideal helical reconstruction of the TM segment was relaxed and equilibrated in a membrane bilayer using MD simulations. These additional two domains were eventually linked to the periplasmic YscD ring structure and assembled in a 24-mer conformation. We used again our flexible docking protocol to combine the low-resolution spatial restraints obtained from the cryo-ET maps and the YscD N-terminal structure to assembly the YscD cytoplasmic 24-mer ring (*Figure 4*).

## Method details for molecular dynamics and modeling

Sequence alignments were performed using the *ClustalWS* algorithm and visualized/rendered with *Jalview 2.7*. *Jpred* 3 and *DisEMBL* were used to perform, respectively, secondary structure and disorder predictions. The software *superpose* (*Krissinel and Henrick, 2004*) from the *CCP4* (*Potterton et al., 2004*) program suite was employed to perform structural alignment based on the matching of structural motifs. Loop modeling was performed using *MODELLER* 9.8 (*Fiser et al., 2003*).

All molecular dynamics simulations were performed on systems assembled using the *psfbgen* module of VMD 1.9 (*Humphrey et al., 1996*). The proteins were embedded in an orthorhombic box of water molecules of suitable size as to allow for a minimum distance of 12 Å between the protein and any of the box faces. The web-based *H++* application (*Gordon et al., 2005*) was used to calculate the pKa of titratable residues; since no relevant discrepancies with respect to tabulated values were found, standard protonation states were assigned. A suitable number of ions (Na+ or Cl−) were randomly placed in the box using the *autoionize* module of VMD as to neutralize the system's net charge. All simulated systems were treated at the molecular mechanics (MM) level using the CHARMM22/CMAP (*MacKerell et al., 1998*; *Mackerell et al., 2004*) force field for protein and ions. The TIP3P model (*Jorgensen et al., 1983*) was employed for water molecules when an explicit solvation was required. Otherwise the effects of the water were accounted for by means of the generalized born (GB) implicit solvent model. The van der Waals interaction cutoff distances were set at 12 Å and long-range electrostatic forces were computed using the particle-mesh Ewald summation method with a grid size of <1 Å.

The program *NAMD 2.8* (*Phillips et al., 2005*) was employed to perform all molecular dynamics simulations. All systems were subjected to 10,000 steps of geometry optimization (steepest descent) followed by a suitable equilibration phase. For all equilibrium simulation, constant temperature (T = 300 K) was enforced using Langevin dynamics with a damping coefficient of 5 ps$^{-1}$, constant pressure (p=1 atm) was enforced through the Nosé-Hoover Langevin piston method with a decay period of 100 fs and a damping time constant of 50 fs. A time step of 2 fs was used throughout. Covalent bonds involving hydrogen atoms were constrained using the RATTLE algorithm.

Normal mode analysis (NMA) (*Bahar et al., 2009*) of selected protein structures and essential dynamics analysis (EDA) of molecular dynamics trajectories were performed using Prody 0.9; (*Bakan et al., 2011*) results were rendered using the *matplotlib* (*Hunter, 2007*) graphical library. In the anisotropic network model (ANM) (*Eyal et al., 2006*) approach the structure is modeled as a network of harmonic oscillators: the nodes identifying the α-carbons and the springs accounting for inter-residue interactions. Essential dynamics analysis was used to extract collective protein motions from the molecular dynamics simulations. Like ANM-NMA, EDA constitutes a powerful method to analyze collective motions in biomolecules. In the EDA approach the collective modes of motions are, however, extracted

from a molecular dynamics trajectory rather than from a single structure, and shape and frequency of modes are obtained through diagonalization of the covariance matrix of the α-carbons motion (*Amadei et al., 1993*).

Macromolecular assemblies were constructed with the newly developed swarm intelligence-based protocol (*Degiacomi and Dal Peraro, 2013*) implemented in a software package called "parallel optmization workbench" (pow^er, available at http://lbm.epfl.ch). Pow^er provides predictions for a multimeric structure arrangement on the basis of structural information about its subunits and experimental measures acting as search restraints (as recently shown for the assembly of aerolysin, *Degiacomi et al., 2013*). In a first step, an ensemble of monomer conformations is generated, typically from molecular dynamics simulations or structural biology experiments; this ensemble is then treated as a database of conformations. The advantage of this approach is that assembly prediction is performed using physically plausible structures and dynamic features are directly included. Upon definition of a list of geometric restraints (fitness) and a specific circular symmetry, a particle swarm optimization (PSO) search subsequently tries to arrange the elements of the conformational database in a multimeric assembly, so that all restraints are respected, and steric clashes avoided (i.e., part of the fitness function has an 9–6 Lennard-Jones-type energy potentials avoiding overlaps of different units). Geometric restraints can be typically provided by low resolution electron density maps or experiments such as cross-linking disulfide scanning, mutagenesis or FRET. If necessary, *POW*^er can assemble a multimer on a given substrate, like here for the case of YscD ring on top of YscJ ring. At PSO search completion, a set of solutions having a good score is usually generated. A smaller set of representative solutions, typically less than 10, which all satisfy the initial restraints, is then obtained by clustering the accepted solutions according to their respective root mean square deviation (RMSD).

## Acknowledgements

We thank B Anderson, R Pantelic, K Goldie, and M Chami for expert technical assistance, S A Müller for extensive comments on the manuscript, M Kuhn for plasmid and strain contribution, K Namba for discussions and Martin Jacquot for support with the supercomputing infrastructure of University of Basel.

## Additional information

### Funding

| Funder | Grant reference number | Author |
| --- | --- | --- |
| Swiss National Science Foundation | 3100AOB-128659 | Stefan Münnich, Andreas Diepold, Guy R Cornelis |
| SystemsX.ch | CINA | Mikhail Kudryashev, Daniel Castaño-Díez, Christopher KE Bleck, Julia Kowal, Henning Stahlberg |
| Swiss National Science Foundation | CRSII3_125110 | Mikhail Kudryashev, Marco Stenta, Stefan Schmelz, Marlise Amstutz, Ulrich Wiesand, Daniel Castaño-Díez, Matteo T Degiacomi, Stefan Münnich, Christopher KE Bleck, Julia Kowal, Andreas Diepold, Dirk W Heinz, Matteo Dal Peraro, Guy R Cornelis, Henning Stahlberg |
| Swiss National Science Foundation | NCCR Nano | Mikhail Kudryashev, Christopher KE Bleck, Julia Kowal, Henning Stahlberg |
| Swiss National Science Foundation | NCCR Struct.Biol. | Julia Kowal |
| Swiss National Science Foundation | NCCR TransCure | Henning Stahlberg |
| Swiss National Science Foundation | 00021_122120 | Matteo T Degiacomi, Matteo Dal Peraro |
| Swiss National Science Foundation | 200020_138013 | Matteo T Degiacomi, Matteo Dal Peraro |

| Funder | Grant reference number | Author |
|---|---|---|
| Swiss National Supercomputing Center (CSCS) | projectID s274 | Mikhail Kudryashev, Daniel Castaño-Díez, Henning Stahlberg |

The funders had no role in study design, data collection and interpretation, or the decision to submit the work for publication.

## Author contributions

MK, Conception and design, Acquisition of data, Analysis and interpretation of data, Drafting or revising the article; MDP, GRC, Conception and design, Analysis and interpretation of data, Drafting or revising the article; MS, SS, MA, UW, MTD, SM, CKEB, JK, AD, Acquisition of data, Analysis and interpretation of data, Drafting or revising the article; DC-D, Analysis and interpretation of data, Drafting or revising the article, Contributed unpublished essential data or reagents; DWH, Conception and design, Analysis and interpretation of data, Drafting or revising the article; HS, Conception and design, Analysis and interpretation of data, Drafting or revising the article

# Additional files

### Supplementary files

• Supplementary file 1. (A) Crystallographic data collection and refinement statistics for YscD$^{150-362}$ and YscD$^{150-347}$ G283P. Due to poor refinement statistics of YscD$^{150-362}$ only coordinates for YscD$^{150-347}$ G283P were deposited at the Protein Data Bank (http://www.pdb.org). (B) *Y. enterocolitica* strains. (C) Plasmids. (D) List of used oligonucleotides.

### Major dataset

The following datasets were generated:

| Author(s) | Year | Dataset title | Dataset ID and/or URL | Database, license, and accessibility information |
|---|---|---|---|---|
| Schmelz S, Wiesand U, Stenta M, Münnich S, Widow U, Cornelis GR, Heinz DW | 2013 | The *Yersinia* T3SS basal body component YscD reveals a different structural periplasmatic domain organization to known homologue PrgH | 4ALZ; http://www.rcsb.org/pdb/explore/explore.do?structureId=4ALZ | Publicly available at the Protein Data Bank (http://www.rcsb.org/pdb/home/home.do). |
| Kudryashev M, Stenta M, Schmelz S, Amstutz M, Wiesand U, Castaño-Díez D, et al. | 2013 | Type III secretion system (Injectisome) of *Yersinia enterocolitica* in situ | EMD-5694; http://www.ebi.ac.uk/pdbe/entry/EMD-5694 | Publicly available at the EMDB (http://emdatabank.org). |

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
