## [Decision Letter]

Thank you for sending your work entitled “In situ structural analysis of the
*Yersinia enterocolitica* injectisome” for consideration at
*eLife*. Your article has been favorably evaluated by a Senior editor,
a Reviewing editor, and 2 reviewers.

The Reviewing editor and the two reviewers discussed their comments before we reached
this decision, and the Reviewing editor has assembled the following comments to help you
prepare a revised submission.

In general, the reviewers are very positive about the interesting and unexpected results
reported in your manuscript. In particular, the possibility to compare in vitro with in
situ results was seen as very interesting. There was only one major technical concern
that should be addressed.

The methods do not rule out that over-alignment has lead to artifacts in the
sub-tomogram averaging structures. The risk of this is high for highly-symmetrized
sub-tomogram averaging structures. The figures in some cases show disconnected and
inverted densities, which can result from such artifacts. This is a particular concern
for the YscC structure in liposomes (Figure 6)
but also for other panels. Further, the substantial differences in structure dependent
on the mask used in Figure 2 could also hint at
this possibility. The authors should rule out this possibility by preferably:

1) aligning two halves of each dataset completely independently (from the start), and
comparing the two halves to calculate resolution and to identify reliable features;

2) Or, if not, then make sure that low-pass filters are applied at each iteration such
that zero information is passed beyond resolutions that are significantly below the
final resolution obtained. The position of YscV can be interpreted only after these
tests.

---

## [Author Response]

We are aware of the significant risk of overfitting in the alignment of noisy data to a
reference. As you point out, classical sub-tomogram averaging may produce such
overfitting, while the “gold standard” procedure reduces this risk by
splitting the dataset into two sub-sets and processes these completely independently.
Unfortunately, applying this algorithm to sub-volume averaging is challenging if only a
low number of sub-volume “particles” is available. We have now
re-processed our injectisome sub-volumes, following the “gold standard”
strategy with our “Dynamo” software. This resulted in a more conservative
resolution estimate of 4 nm for our final injectisome structure. We now describe this
“gold standard” processing approach in an additional section within the
Materials and methods section and a panel in Figure 2—figure supplement 1, and we have updated the manuscript
accordingly.

The new, more reliable processing, however, did not affect our findings or the
interpretation of the results: the positions of the rings in the injectisome structure,
including the putative YscV ring, remained as before, and significant variations in the
injectisome lengths after classification and averaging are observed as previously
described. These dimensions are also supported by intermembrane distances measured at
individual single injectisomes that we also show in Figure 2.

As for the YscC structure, due to the small number of particles we could not achieve a
reliable convergence with only half of the particles. The average from all available
YscC sub-volumes, however, shows similar dimensions as the individual YscC sub-volumes
showed before averaging, which leads us to believe that the sub-volume averaging reports
a valid average structure within the specified resolution limits. To provide a reliable
estimate of this resolution, we now used the more conservative threshold for the
resolution (FSC=0.5), reporting a resolution of 3 nm. We have updated Figure 6 and the manuscript accordingly.